# Effects of *Cymatocarpus solearis* (Trematoda: Brachycoeliidae) on its second intermediate host, the Caribbean spiny lobster *Panulirus argus*

Tomás Franco-Bodek[1], Cecilia Barradas-Ortiz[2], Fernando Negrete-Soto[2], Rossanna Rodríguez-Canul[3], Enrique Lozano-Álvarez[2], Patricia Briones-Fourzán[2]*

**1** Posgrado en Ciencias del Mar y Limnología, Universidad Nacional Autónoma de México, Ciudad Universitaria, Ciudad de México, Mexico, **2** Unidad Académica de Sistemas Arrecifales, Instituto de Ciencias del Mar y Limnología, Universidad Nacional Autónoma de México, Puerto Morelos, Quintana Roo, Mexico, **3** Laboratorio de Inmunología y Biología Molecular, Departamento de Recursos del Mar, Centro de Investigación y de Estudios Avanzados del Instituto Politécnico Nacional-Unidad Mérida, Mérida, Yucatán, Mexico

* briones@cmarl.unam.mx

## Abstract

Many digenean trematodes require three hosts to complete their life cycle. For *Cymatocarpus solearis* (Brachycoeliidae), the first intermediate host is unknown; the Caribbean spiny lobster *Panulirus argus* is a second intermediate host, and the loggerhead turtle *Caretta caretta*, a lobster predator, is the definitive host. Trophically-transmitted parasites may alter the behavior or general condition of intermediate hosts in ways that increase the hosts' rates of consumption by definitive hosts. Here, we examined the effects of infection by *C. solearis* on *P. argus* by comparing several physiological and behavioral variables among uninfected lobsters (0 cysts) and lobsters with light (1–10 cysts), moderate (11–30 cysts), and heavy (>30 cysts) infections. Physiological variables were hepatosomatic index, growth rate, hemocyte count, concentration in hemolymph of cholesterol, protein, albumin, glucose, dopamine (DA) and serotonin (5-HT). Behavioral variables included seven components of the escape response (delay to escape, duration of swimming bout, distance traveled in a swimming bout, swim velocity, acceleration, force exerted, and work performed while swimming). There was no relationship between lobster size or sex and number of cysts. Significant differences among the four lobster groups occurred only in concentration of glucose (lower in heavily infected lobsters) and 5-HT (higher in heavily and moderately infected lobsters) in plasma. As changes in 5-HT concentration can modify the host's activity patterns or choice of microhabitat, our results suggest that infection with *C. solearis* may alter the behavior of spiny lobsters, potentially increasing the likelihood of trophic transmission of the parasite to the definitive host.

**Data Availability Statement:** All relevant data are within the paper and its Supporting Information files.

**Funding:** This study received funding from Universidad Nacional Autónoma de México (Program UNAM-DGAPA-PAPIIT, project IN206117 (https://dgapa.unam.mx/), granted to P. B.-F. The Consejo Nacional de Ciencia y Tecnología (CONACYT-México) (https://conacyt.mx/) provided a Master's scholarship (2018-000012-01NACF-08432) for T.F.-B. The funders had no role in study design, data collection and analysis, decision to publish, or preparation of the manuscript.

**Competing interests:** The authors have declared that no competing interests exist.

# Introduction

Parasitism is an important factor influencing the composition and structure of populations and communities [1, 2]. Parasites can affect many phenotypical characteristics of their hosts, such as growth rate, reproductive rate, nutritional condition, fecundity, immune response, and concentration in the hemolymph of metabolites or neuromodulators, among others [3–5]. Alterations in neuromodulators may in turn cause behavioral changes that increase susceptibility of hosts to predation [4, 6–8].

Digenean trematodes (Platyhelminthes) are widely distributed parasites that usually require three hosts to complete their life cycle: a definitive host and two intermediate hosts [9]. Trematode eggs released in the definitive host's feces are ingested by the first intermediate host, which is usually a gastropod [10]. Eventually, cercariae are produced in the body of the first intermediate host and released into the water. Upon finding a second intermediate host, the cercariae penetrate its body and migrate to the appropriate tissue, where they encyst, becoming metacercariae. Metacercariae infect the definitive host via consumption of the second intermediate host [10].

Trophically-transmitted parasites may alter the behavior, appearance, or general condition of intermediate hosts in ways that increase their rates of consumption by predatory definitive hosts (review in [6]) and, in some cases, the effects may be multidimensional [4, 7]. It is important to determine which traits of a host are affected by a parasite because, even if the effects are subtle, they could change the host population dynamics and, by extension, the communities where they live [11, 12]. Trophically-transmitted parasites tend to have more dire consequences for the intermediate hosts than for the definitive hosts because the intermediate host needs to be eaten for the parasite to complete its life cycle in a definitive host, where it attains sexual maturity and reproduces [13].

A wide variety of trematodes use crustaceans as second intermediate hosts [14, 15]. This is the case for *Cymatocarpus solearis* (Brachycoeliidae) [16]. Definitive hosts for *C. solearis* are marine turtles, particularly loggerhead turtles *Caretta caretta* [17, 18]. The first intermediate host for *C. solearis* has not been identified yet but, to date, three species of decapod crustaceans have been reported as second intermediate hosts: the hermit crab *Dardanus tinctor* (formerly *Pagurus tinctor*) in the Persian Gulf [19], the channel crab *Maguimithrax spinosissimus* (formerly *Mithrax spinosissimus*) in Cuba [20], and the Caribbean spiny lobster, *Panulirus argus*, in the Mexican Caribbean and in Cuba [20–23].

Caribbean spiny lobsters constitute one of the most valuable fishing resources throughout the wider Caribbean region. Following a protracted larval phase that develops in oceanic waters, the postlarvae of *P. argus* settle in shallow marine vegetation habitats, where the juveniles remain for a few months. Subadults migrate to coral reef habitats where the adults live, although the latter may use a variety of habitats from patch reefs in shallow reef lagoons to deep reefs and rocky bottoms about 80 m in depth [24]. Caribbean spiny lobsters consume a variety of small marine invertebrates (mollusks, crustaceans, echinoderms, worms) and are consumed by many predators such as groupers, snappers, nurse sharks, stingrays, triggerfish, octopuses, dolphins, and loggerhead turtles [25, 26]. These lobsters are highly gregarious, with multiple individuals often sharing a single shelter. The social behavior of *P. argus* is mediated by conspecific chemical communication and is capitalized by some fisheries that are based on the extensive use of artificial shelters (casitas) that can harbor many lobsters [27].

Bahía de la Ascensión is a large bay in the Caribbean coast of Mexico with a successful casita-based fishery for spiny lobsters and where infection by *C. solearis* in *P. argus* was first discovered. Cysts of *C. solearis* appear as whitish spheres, ~1 mm in diameter, embedded in the muscles of infected lobsters. Cysts in the abdominal muscles are visible to the naked eye

through the translucid membrane between the cephalothorax and abdomen, and the cuticle along the ventral wall of the abdomen [21]. Rapid field surveys based on visual assessment yielded estimates of prevalence of infection in the lobster population of the bay of 21% in 2011 [22] and 14% in 2016 [23], with the probability of infection increasing with size of lobsters. However, Cruz-Quintana [20] and Briones-Fourzán et al. [22] fully dissected infected lobsters in the laboratory and found more metacercariae in the muscles of the cephalothorax and the coxae than in the abdominal muscles (Fig 1), suggesting that visual assessment underestimates the prevalence of *C. solearis* infection in lobster populations, but whether and how *C. solearis* affects *P. argus* remained to be determined. In crustacean hosts, parasites may affect the nutritional condition, blood chemistry, or concentration of hemocytes in the hemolymph [28], the escape response [29–31], or the concentration of certain biogenic amines (neurotransmitters, neuromodulators, and neurohormones), causing behavioral changes that may increase their vulnerability to predators [6, 32]. Therefore, in species of commercial value, such as spiny lobsters, parasites may indirectly affect the fisheries if infected lobsters are more susceptible to predation [11].

In the present study, we assessed the effects of infection by *C. solearis* on several biochemical and physiological traits of lobsters. We examined whether infection affected the swimming performance of lobsters during the escape response, which consists of swimming backwards as a result of rapid abdominal flexions ("tail flips") [33]. Because in many infections the energy normally used by the host to grow is diverted to combat the infection or to compensate for the problems caused by the infection [5, 34], we examined the effects of infection on growth rate and nutritional condition of lobsters. We also investigated whether infection with *C. solearis* altered the concentration of two biogenic amines in the host's plasma: dopamine (DA), which is involved in gonad maturation, carbohydrate metabolism, and learning processes, and serotonin (5-HT), which modulates response to stress, sexual and agonistic behavior, physiological process (including control of energy balance), circadian rhythms, and neurogenesis [35–38]. We did this because changes in the concentration of biogenic amines may alter the host's activity patterns or choice of microhabitat, potentially increasing exposure to predators [6, 8, 32, 39]. In crustaceans, the concentration of hemocytes in the hemolymph may increase as part of the immune response [40] or decrease as a result of heavy infections [41, 42], although the hemocyte counts may increase or decrease in the same host depending on the type of parasite [43–45]; therefore, we examined this trait in the *P. argus-C. solearis* system. Finally, we examined the relationship between number of parasites and host size and sex, and estimated sensitivity and specificity of visual assessment of infection in live lobsters versus assessment by full host dissection.

## Materials and methods

### Experimental lobsters

The study was conducted throughout 2018 and 2019. Live, legal-sized lobsters (~≥74 mm carapace length, CL) were purchased from the fishing cooperatives that catch lobsters from casitas in Bahía de la Ascensión (centered at 19˚ 40' N, 87˚ 33' W) and Bahía Espíritu Santo (centered at 19˚ 20' N, 87˚ 34' W), located on the Caribbean coast of Mexico. The lobsters were held in seawater tanks, 3 m- and 2 m-diameter and 1 m in height, with a water level of 80 cm, in UNAM's facilities at Puerto Morelos (20˚52' N, 86˚ 52' W). The tanks had an open seawater flow, pumped from the Puerto Morelos reef lagoon. Therefore, temperature and salinity were similar to ambient conditions (water temperature range for 2018–2019: 24.9–31.1˚C; salinity range: 34.47–36.37 psu [46]). The lobsters were fed in excess three times a week with frozen (thawed) mussels and crabs. Any food remains were siphoned out the following day.

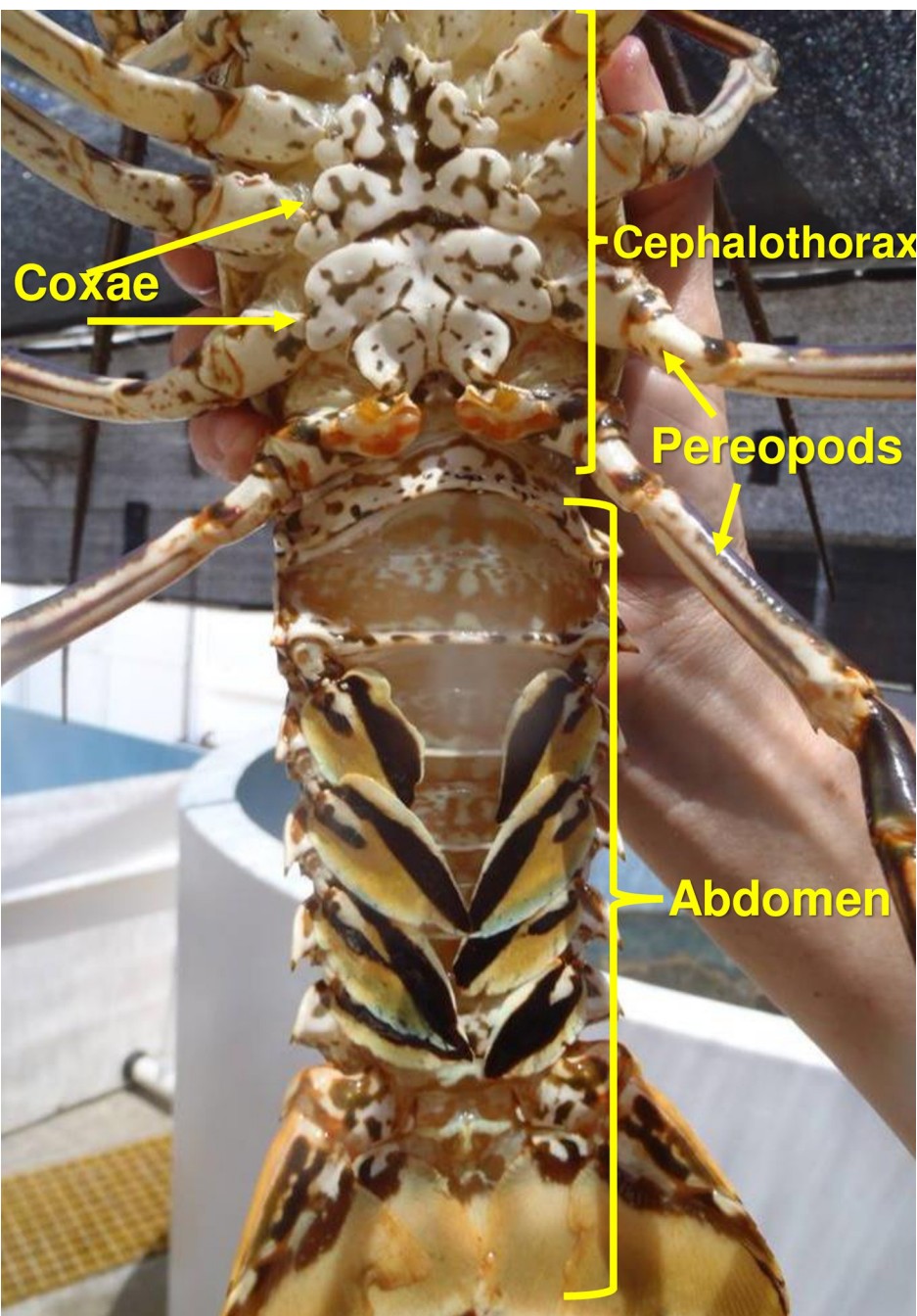

**Fig 1. Ventral view of a male *Panulirus argus*.** Anatomical parts of a spiny lobster mentioned in the text. The photo is of a large male. The body is composed of two main parts, cephalothorax (i.e., the fusion of head and thorax) and abdomen. The pereopods are the walking legs. The coxae are the first segments of the pereopods joining the legs to the thorax. Photo credit: F. Negrete-Soto.

After two weeks of acclimatization, each lobster was measured with Vernier calipers (CL, from between the supraorbital horns to the posterior end of the cephalothorax). A digital photograph of the carapace was then taken and assigned a number. The photographs allowed post-molt identification of lobsters, because the body markings and patterns in the carapace of

spiny lobsters are conserved after molting [47, 48]. Some lobsters ($n = 107$) were maintained for several weeks to months to estimate their growth rate; some ($n = 80$) were used to measure swimming performance (see below). The latter were returned to their holding tanks for several days to weeks prior to measuring the rest of the variables.

A sample of hemolymph was extracted from each lobster to estimate hemocyte density ($n = 82$) and the concentration of metabolites and biogenic amines (see below). The lobsters were measured again and weighed on a digital scale. The distal third of one pleopod was excised to determine the molt stage by observation under a microscope (see [49]), and the lobsters were euthanized by lowering their metabolism in iced seawater. Then the cephalothorax and abdomen were separated; the hepatopancreas was extracted, blotted to remove excess water, and weighed. The body parts of each lobster were completely dissected and examined for cysts of *C. solearis*, which are easily detected to the naked eye as small whitish spheres (~1 mm in diameter) embedded in muscle tissue [20, 21]. All cysts in each lobster were counted and a few were haphazardly selected to extract and examine the metacercariae under the microscope to ensure their identity. In all cases, the excysted metacercariae corresponded to the description of *C. solearis* [21].

## Ethics statement

All experimental lobsters were purchased from registered Fishing Cooperatives, were maintained in good conditions until examination, and were euthanized quickly and humanely.

## Comparison of lobster characteristics

**Swimming performance (escape response).** To determine whether infection with *C. solearis* affects the swimming performance of *P. argus*, we measured three variables based on the protocol designed by Briones-Fourzán et al. [50]: (1) delay (s) to escape (how long before the lobster initiated the escape response), (2) duration of a swimming bout (s), i.e. the series of full tail flips between stimulation and rest, and (3) distance (m) traveled in a swimming bout. With these data, we further estimated (4) overall swimming velocity (m/s), (5) acceleration (m/s$^2$), (6) force exerted (body weight in kg × acceleration, in newtons, N), and (7) work performed while swimming (acceleration × force, in joules, J) [50].

We used a concrete channel, 5 m long and 40 cm wide with a 50 cm water depth, marked every centimeter. A partition with a pulley formed a $50 \times 60$ cm "start compartment" at one end of the channel. One lobster was introduced into the compartment, which was covered to reduce disturbance. After the lobster stabilized for 2–3 min, a person would rapidly introduce the hands into the compartment and try to grab the lobster to simulate a predator attack while the partition of the compartment was lifted, allowing the lobster to swim backwards along the channel [50]. Two observers operating stopwatches were positioned one at each side of the channel. One of them measured the time elapsed from the simulation of the predator attach to the end of the swimming bout (Time A), while the other one measured the time elapsed between the moment the lobster exited the compartment and the end of the swimming bout (Time B). The delay to escape was estimated as Time A–Time B. Experimental runs in which a lobster did not swim (e.g., walked) or attempted to escape in the direction opposite to the wooden partition were not considered. In total, we used data from 78 lobsters used in this experiment, all of them in intermolt. Ovigerous females were excluded.

**Growth rate.** To measure the growth rate of lobsters, the holding tanks were checked daily for molts, which were extracted and contrasted with the photographs taken when the lobsters were first measured ($CL_1$) to identify the individual that molted. About 20 days later, once the exoskeleton had completely hardened, the molted individual was measured again

(CL$_2$). The weekly growth rate was estimated as CL$_2$ –CL$_1$/number of days elapsed between molts × 7 [51].

**Concentration of biogenic amines.** To determine the concentration of DA and 5-HT in the lobster's hemolymph, 1 ml of hemolymph was extracted and dissolved 2:1 in LIS to avoid clotting [52]. LIS is a "lobster isotonic solution" that uses EDTA (ethylene-diamine-tetracetic acid) as an anticoagulant, based on its chelant properties (NaCl 350 mM, KCl 10 mM, buffer HEPES 10 mM, EDTA-Na 10 mM, pH 7.3, approximate osmolarity 790 mOsm/kg). The hemolymph was extracted with a 3 ml syringe containing LIS, inserted at the basis of the fifth pereiopods. The hemolymph was immediately centrifuged at 4000 g to separate the plasma, which was frozen at -20˚ C until further processing.

The samples were processed using the ALPCO Dopamine quantification kit (17-DOPHU-E01.1) and the ALPCO DRG Serotonin FAST ELISA quantification kit (EIA-5061), following the manufacturer's protocols. Absorbance readings for both neuromodulators were taken at 25˚ C at 450 nm (650 nm reference wavelength). Quantification of known samples (as μg ml$^{-1}$) was achieved by comparing their absorbance with a reference curve prepared with known standard concentrations [32].

**Nutritional condition: Concentration of metabolites.** To determine whether infection with *C. solearis* affected nutritional condition of lobsters, we measured the concentration in hemolymph of total protein, cholesterol, glucose, and albumin. SPINREACT colorimetric kits (Glucose-LQ 41010, Albumin 1001020, Total Protein 1001291, Cholesterol-LQ 41020) were used, following the manufacturer protocols with slight modifications. Because the readings were done in a spectrophotometer using 96-well plates, the protocol was modified using 200 μl of the kit's reagent and 10 μl of the hemolymph sample dissolved in LIS. The cholesterol readings were measured at 505 nm after incubating the reaction for 5 min at 37˚ C. The glucose readings were measured at 505 nm after incubating the reaction for 10 min at 37˚ C according to the kit's specifications. The albumin readings were measured at 630 nm, whereas the total protein readings were measured at 540 nm without incubation. As the concentration of metabolites in the hemolymph of crustaceans is influenced by molt stage [5], only samples from individuals in intermolt were used in these analyses.

**Nutritional condition: Hepatosomatic index.** The hepatosomatic index (HI) provides an indication of the nutritional condition of lobsters, including *P. argus* [53, 54]. The HI was estimated as wet weight of hepatopancreas/lobster weight × 100.

**Total hemocyte count.** The concentration of hemocytes in the hemolymph was estimated by resuspending in LIS the cellular pellet obtained upon extracting the plasma. An aliquot of 20 μl from the resuspended cellular pellet was put in a Neubauer chamber where the hemocytes were counted [55]. The following equation was used to estimate the density of hemocytes: (total count of hemocytes/8) × dilution factor × 1000, where 8 is the number of quadrants in the Neubauer chamber. The dilution factor varied between 1 and 4 depending on the sample. Only lobsters in intermolt were used for this estimation.

## Statistical analyses

The effects of a parasite may vary with infection intensity [56–58]. Exploratory analyses of our results showed a wide dispersion in the results of almost all variables measured in infected individuals. Therefore, lobsters were categorized into four arbitrary groups based on the total number of cysts in their bodies as assessed after full dissection (i.e., after all variables had been measured): uninfected (0 cysts), lightly infected (1–10 cysts), moderately infected (11–30 cysts), and heavily infected (>30 cysts) [29]. This categorization rendered similar samples sizes for all groups and variables, and the mean size and weight of lobsters did not differ

significantly among groups (see Results). However, as we measured some variables in some lobsters and other variables in other lobsters (see S1 Appendix), different analyses were done for different groups of variables. Two separate one-factor multivariate analyses of variance (MANOVA) with a general lineal model (GLM) approach [59] were used to test for a potential effect of parasite intensity (with four levels: uninfected, lightly infected, moderately infected, and heavily infected) on a combination of the four metabolites (albumin, cholesterol, glucose, and total protein) measured in the hemolymph of lobsters on the one side, and on a combination of the seven components of the escape response of lobsters on the other side. MANOVA results were followed by univariate analyses to examine the individual response variables. Data on the concentration of metabolites met all MANOVA assumptions, whereas in the case of the escape response variables, the assumption of homogeneity of the variance-covariance matrix was not met. However, departure from this assumption should not be an issue if sample sizes are equal or nearly equal [59, 60], and sample sizes were 19–21 for this particular group of variables.

To test for potential effects of parasite intensity on the remaining variables (concentration of dopamine, concentration of serotonin, hemocyte count, hepatosomatic index, and growth rate), data from each variable were individually subjected to an analysis of covariance (ANCOVA, also with a GLM approach), with parasite intensity as the factor (with four levels) and lobster size (CL) as the covariate. Significant results of GLMs were followed by Tukey's HSD multiple comparisons test. Statistical analyses were done with the software Statistica v.10 (StatSoft, Inc., Tulsa, OK, USA).

**Relationship between number of metacercariae and lobster size.** Metacercariae may remain in a host's body throughout its life, but their presence does not preclude the host from further reinfections. If parasites accumulate in a host as it grows without compromising its survival, then a positive linear relationship between the number of parasites and the size of the host would be expected, whereas if host mortality increases with number of parasites, then the relationship would be negative [58]. A non-linear relationship may indicate that the number of parasites increases with host size only to a point but then decreases due to the loss of highly parasitized individuals from the host population [3, 58, 61]. Also, in some decapods, intensity of infection varies with sex [58]. To examine whether number of cysts depended on lobster size and sex, we transformed the data to Log (number of cysts + 1) [58] and subjected these data to an ANCOVA, with sex as a categorical factor and CL as the covariate.

**Validity measures for the visual assessment of infection by *C. solearis*.** To date, prevalence of infection by *C. solearis* in wild lobsters has been estimated only by visual assessment, searching carefully for cysts in the abdominal muscles through the arthrodial membranes and the abdominal ventral wall [21–23]. If visual assessment provided a good approximation to real prevalence, it could constitute a rapid, non-invasive way of assessing infection in wild lobster populations, which would be of interest to both researchers and fishers. However, cysts may occur in muscles in the cephalothorax that are not evident to the naked eye [20, 22]. Therefore, visual assessment of infection needs to be validated by comparing it with a more precise test ("gold standard").

Validity measures of a test are based on two criteria: sensitivity and specificity [62]. Sensitivity measures the proportion of correctly identified positives (e.g., the proportion of infected lobsters), whereas specificity measures the proportion of correctly identified negatives. For our purposes, the gold standard was the determination of the presence of metacercariae after full dissection of the entire lobster. Sensitivity of visual assessment was estimated as the number of lobsters classified as infected by visual inspection/number of lobsters classified as infected after full dissection. Specificity of the visual test was estimated as the number of lobsters without cysts/number of lobsters without cysts plus the number of false positive identifications [62].

We estimated 95% confidence intervals for proportions using the Wilson method with continuity correction [63].

## Results

### Size and weight of lobsters

The size range of lobsters was 72.8–142.5 mm CL. After separating the lobsters in four groups based on the presence of metacercariae and intensity of infection, no significant differences were found among the four groups of lobsters in either size ($F_{3,122}$ = 0.245, $p$ = 0.865) (Fig 2A) or weight ($F_{3,119}$ = 0.235, $p$ = 0.872) (Fig 2B). Means and 95% confidence intervals of all response variables of the four groups of lobsters are given in S1 Table.

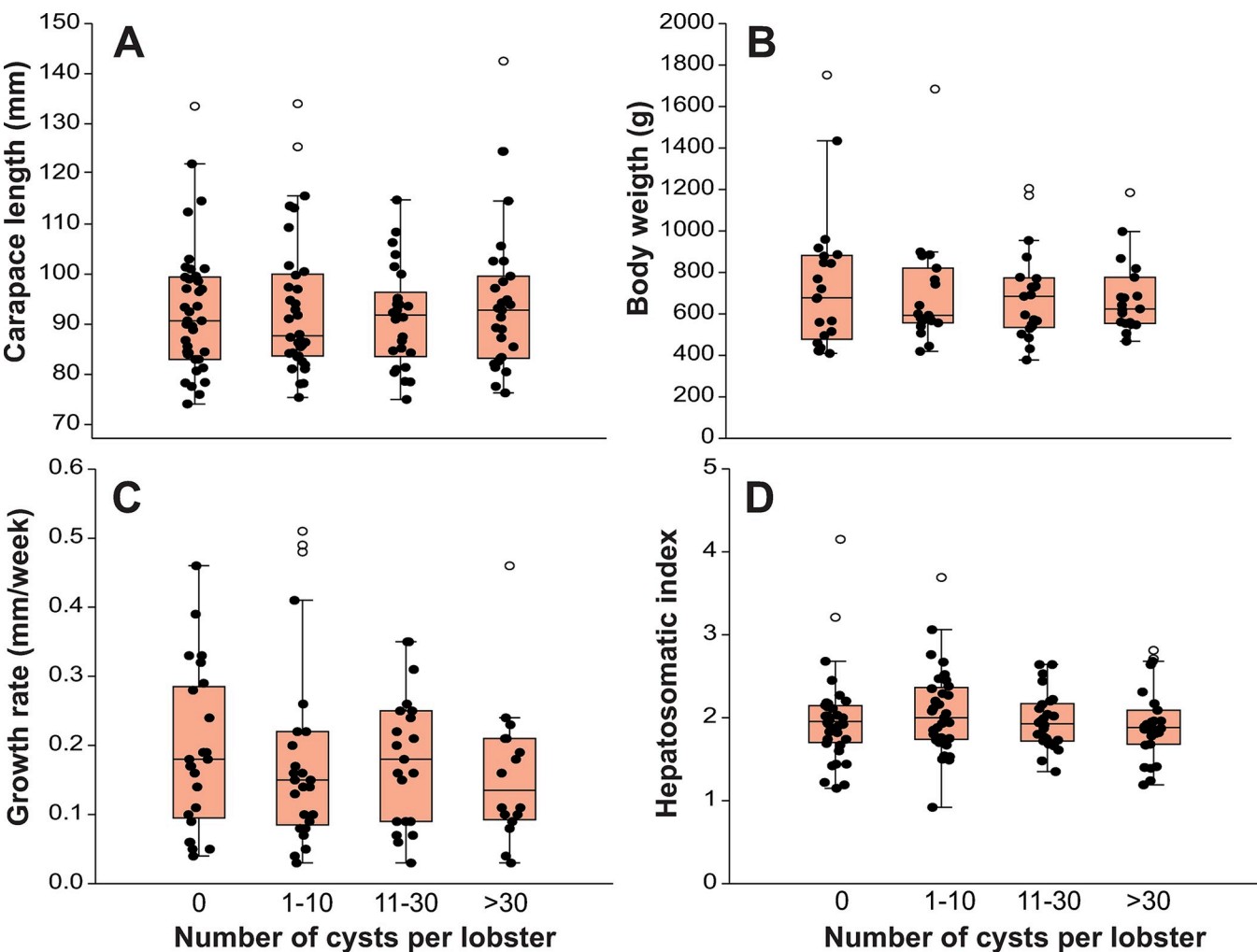

**Fig 2. Body dimensions, growth rate, and hepatosomatic index of lobsters.** Box plots of (A) carapace length (mm), (B) weight (g), (C) growth rate (mm/week), and (D) hepatosomatic index in *Panulirus argus* lobsters categorized into four grades of infection by metacercariae of *Cymatocarpus solearis*: uninfected (0 metacercariae), lightly infected (1–10 metacercariae), moderately infected (11–30 metacercariae), and heavily infected (>30 metacercariae). Central lines in boxplots correspond to medians, box extremities indicate interquartile range (IQR, 1st and 3rd quartiles), whiskers include all data within 1.5 times the IQR, white dots outside the whiskers denote outliers.

## Growth rate and hepatosomatic index

Neither size nor parasite intensity significantly affected the weekly growth rate of lobsters (Table 1, Fig 2C). The hepatosomatic index (HI) was significantly affected by lobster size, but not by parasite intensity (Table 1, Fig 2D). HI tended to decrease with increasing lobster size, but the relationship explained only 7% of the data variation.

## Swimming performance (escape response)

The presence and intensity of trematodes had no significant effects on the combination of escape response variables of individual lobsters (Wilk's lambda = 0.753, $F$ = 1.412; df = 21, 197; $p$ = 0.504). All univariate tests confirmed a non-significant effect on each of the separate components ($p$-values: 0115–0.692) (Fig 3).

## Concentration of metabolites in hemolymph

The presence and intensity of trematodes had no significant effects on the combination of concentrations of albumin, glucose, cholesterol, and total protein in plasma of individual lobsters (Wilk's lambda = 0.806, $F$ = 1.412; df = 12, 161; $p$ = 0.330). The univariate tests confirmed a non-significant effect on the concentration of albumin, cholesterol, and total protein ($p$-values: 0.518–0.738) (Fig 4A, 4B and 4E), but revealed a significant effect of trematode intensity on concentration of glucose ($F$ = 3.388; df = 3, 66; $p$ = 0.023; power: 0.73) (Fig 4C).

**Table 1. Effects of lobster size and parasite intensity on traits of *Panulirus argus*.**

| Variable | Effect | df | MS | F | p |
|---|---|---|---|---|---|
| Growth rate (mm/week) | Intercept | 1 | 0.140 | 8.963 | 0.003 |
| | Size | 1 | 0.029 | 1.888 | 0.172 |
| | Parasite intensity | 3 | 0.015 | 0.982 | 0.405 |
| | Error | 102 | 0.016 | | |
| Hepatosomatic index (HI) | Intercept | 1 | 19.328 | 90.886 | <0.001 |
| | Size | 1 | 1.881 | 8.847 | 0.004 |
| | Parasite intensity | 3 | 0.117 | 0.549 | 0.650 |
| | Error | 117 | 0.213 | | |
| Dopamine (ng/ml) | Intercept | 1 | 324133.4 | 0.351 | 0.556 |
| | Size | 1 | 12599.6 | 0.014 | 0.907 |
| | Parasite intensity | 3 | 281380.5 | 0.305 | 0.822 |
| | Error | 64 | 923589.6 | | |
| Serotonin (ng/ml) | Intercept | 1 | 8362.56 | 4.613 | 0.035 |
| | Size | 1 | 340.95 | 0.188 | 0.666 |
| | Parasite intensity | 3 | 20954.58 | 11.559 | <0.001 |
| | Error | 78 | 1812.83 | | |
| Hemocyte count ($n$/ml) | Intercept | 1 | 2.46E+13 | 10.977 | 0.001 |
| | Size | 1 | 2.08E+12 | 0.931 | 0.338 |
| | Parasite intensity | 3 | 8.68E+11 | 0.388 | 0.762 |
| | Error | 77 | 2.24E+12 | | |

Results of GLMs (α = 0.05) on data of several variables compared between four groups of lobsters differing in parasite intensity: uninfected (0 cysts), lightly infected (1–10 cysts), moderately infected (11–30 cysts), and heavily infected (>30 cysts), using lobster size as a covariate. *N* lobsters used for each variable: dopamine: 14–22 per group; serotonin: 20–22 per group; HI: 26–36 per group; growth rate: 16–25 per group; hemocyte count: 18–22 per group.

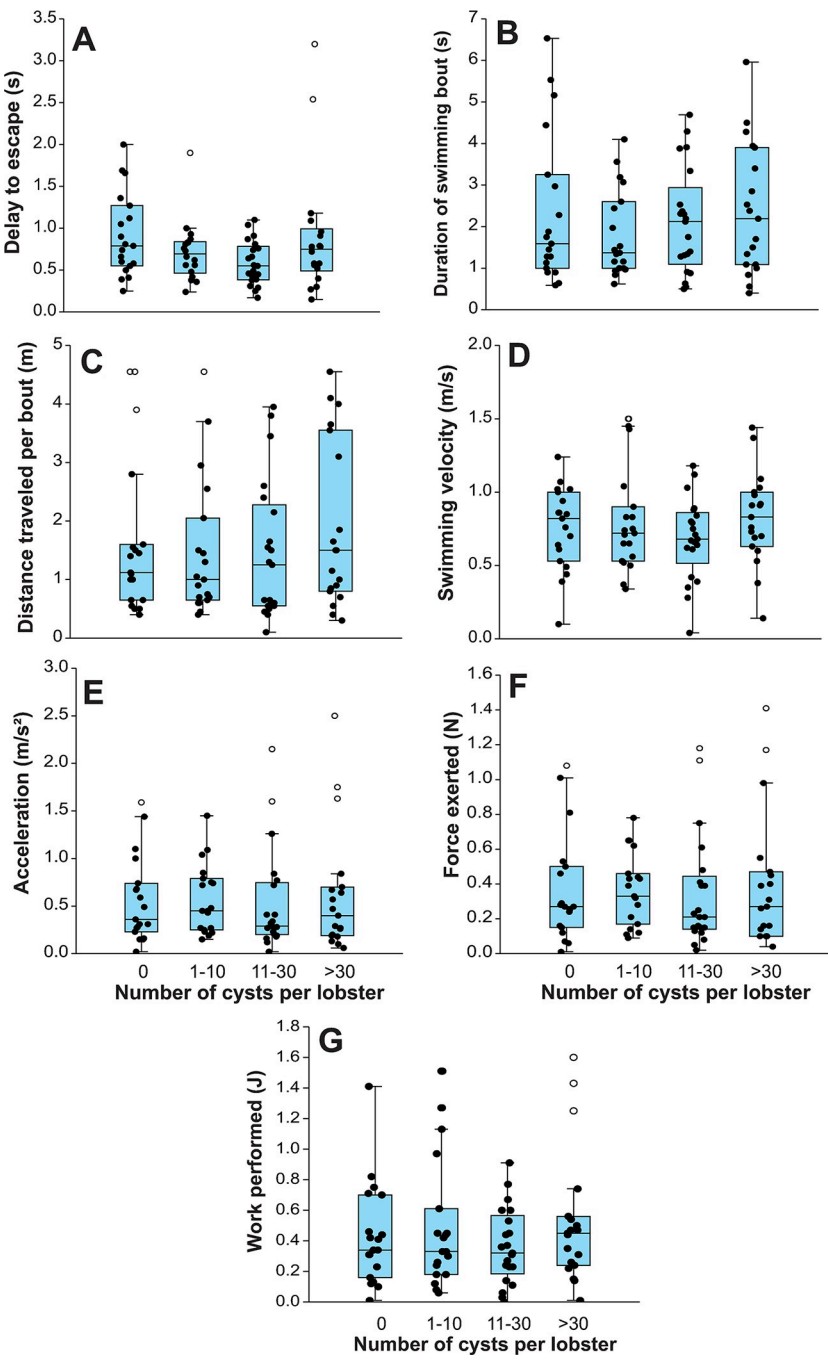

**Fig 3. Escape response of lobsters.** Box plots of seven components of the swimming performance during the escape response: (A) delay to escape (s), (B) duration of swimming bout (s), (C) distance travelled during bout (m), (D) swimming velocity (m/s), (E) acceleration (m/s²), (F) force exerted (N), and (G) work performed (J), in lobsters *Panulirus argus* categorized into four grades of infection by metacercariae of *Cymatocarpus solearis*: uninfected (0 metacercariae), lightly infected (1–10 metacercariae), moderately infected (11–30 metacercariae), and heavily infected (>30 metacercariae). Central lines in boxplots correspond to medians, box extremities indicate interquartile range (IQR, 1st and 3rd quartiles), whiskers include all data within 1.5 times the IQR, white dots outside the whiskers denote outliers.

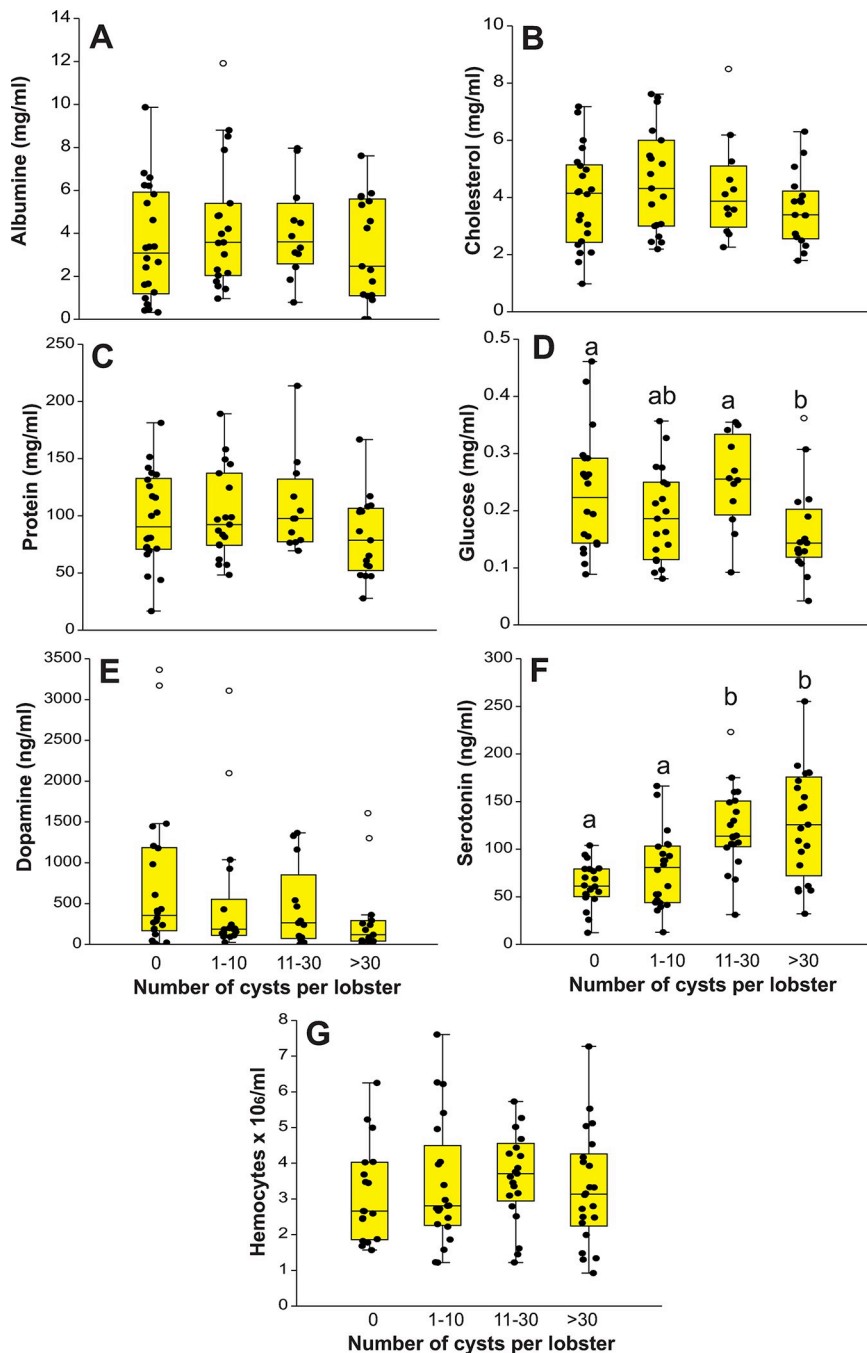

**Fig 4. Components in hemolymph of lobsters.** Box plots of concentration of components in the hemolymph of *Panulirus argus* lobsters categorized into four grades of infection by metacercariae of *Cymatocarpus solearis*: uninfected (0 metacercariae), lightly infected (1–10 metacercariae), moderately infected (11–30 metacercariae), and heavily infected (>30 metacercariae). (A) albumine (mg/ml), (B) cholesterol (mg/ml), (C) protein (mg/ml) (D) glucose (mg/ml), (E) dopamine (ng/ml), (F) serotonine (ng/ml), and (G) hemocytes (N × 10⁶/ml). In (D) and (F), different letters above bars denote different means. Central lines in boxplots correspond to medians, box extremities indicate interquartile range (IQR, 1st and 3rd quartiles), whiskers include all data within 1.5 times the IQR, white dots outside whiskers denote outliers.

## Concentration of DA and 5-HT in hemolymph and hemocyte count

Lobster size and parasite intensity had no significant effects on concentration of DA in plasma (Table 1, Fig 4E). Concentration of 5-HT differed significantly among groups of lobsters ($p < 0.001$), with no significant effect of lobster size (Table 1) Values of serotonin in plasma were higher in heavily and moderately infected lobsters than in lightly infected and uninfected lobsters (Fig 4F). Total hemocyte count was not significantly affected either by lobster size or parasite intensity (Table 1, Fig 4G).

## Validity measures of visual assessment of *C. solearis*

Of 208 lobsters examined, 28 (13.5%) had at least one visible cyst. However, upon full dissection, cysts were found in 110 lobsters (52.9%) (Table 2). Therefore, sensitivity of visual assessment was 25.4% {95% CI: 17.6%, 33.2%} (28 visually identified as infected/110 identified as infected by dissection). This result indicates that visual assessment detected about one fourth (25%) of all lobsters infected with metacercariae of *C. solearis*, i.e., that for each lobster visually identified as infected, there were three other lobsters also infected but with cysts in muscles not visible to the naked eye. Of the 98 lobsters that proved to have no cysts after full dissection, one had been erroneously identified as infected by visual inspection. Therefore, specificity of visual assessment was 99.1% {94.5%, 99.9%} (98 non-infected, but 1 false positive) (Table 2). The false positive was an individual in which the only visually detected "cyst" turned out to be a small scar in the muscle.

## Number of cysts vs lobster size

In the 208 lobsters examined the total numbers of cysts ranged from 0 to 226, with a median abundance of 1 (IQR: 0–16.3). Considering only lobsters with 1 or more cysts (N = 110), the median intensity was 13.5 (3.3–41.0). Infected lobsters had less cysts in the abdomen (median: 2 (0–5.5), range: 0–32) than in the muscles of the coxae and cephalothorax (median: 12 (3–33.8), range: 1–199). The numbers of lobsters with 0 or $\geq 1$ cysts in abdominal muscles were 128 and 80, respectively, whereas the numbers of lobsters with 0 or $\geq 1$ cysts in cephalothoracic muscles were 101 and 107, respectively. These numbers differed significantly (Yates-corrected $\chi^2_1 = 6.57$, $p = 0.01$). On average, 16% of all cysts observed in all lobsters were in the abdomen and 84% in the cephalothorax. Results of ANCOVA revealed no significant effect of either size ($F_{1, 205} = 0.566$, $p = 0.453$) or sex ($F_{1, 205} = 2.121$, $p = 0.147$) on the number of cysts of *C. solearis* in lobsters (Fig 5).

## Discussion

The present study is the first examining effects of infection by *C. solearis* on *P. argus* by comparing several physiological and behavioral variables among groups of uninfected lobsters and

**Table 2. Validity measures of visual assessment of infection.**

| Visual assessment | Full dissection (Reference test) | | Total |
|---|---|---|---|
| | Negative | Positive | |
| Negative | 97 (true negatives) | 82 (false negatives) | 179 |
| Positive | 1 (false positives) | 28 (true positives) | 29 |
| Total | 98 | 110 | 208 |

Relationship between results of visual assessment of presence of cysts of *Cymatocarpus solearis* infection and full dissection (reference test)

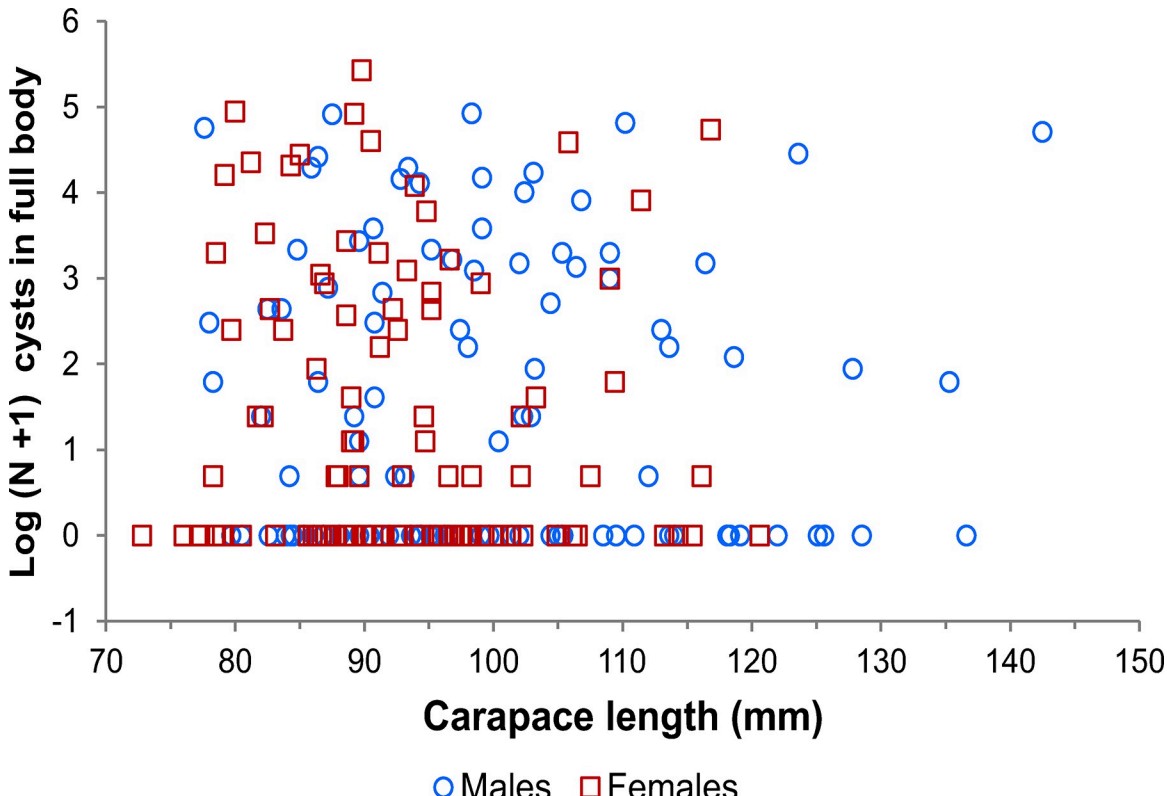

**Fig 5. Lobster size versus number of parasites.** Relationship between size (carapace length, mm) of male and female *Panulirus argus* lobsters and Log (*N* + 1) of *Cymatocarpus solearis* cysts.

lobsters in three different grades of infection (light, moderate, and heavy) based on the parasite load.

Overall, the trematode appears to have little pathological effects on lobsters. The condition of lobsters was generally not affected by the parasite, as the hepatopancreatic index and the concentration of cholesterol, protein, and albumin did not vary with grade of infection. The exception was the concentration of glucose, which was lower in heavily infected lobsters (i.e., those with $\geq$ 30 cysts) than in uninfected lobsters and lobsters with light and moderate infections. Muscle and connective tissue are rich in glucose, which serves as a primary molecule for energy exchange at the cellular level [64]. Cyst walls may be permeable to certain substances, including glucose [65, 66], and although the metabolism of metacercariae is not well understood [67], in some cases they appear to draw nutrition from the host's glucose [7, 68]. This may be the case in the *P. argus-C. solearis* system, although the effect was relatively minor, suggesting that a potential metabolic cost would appear to occur only in severely infected lobsters.

Some parasites affect the swimming performance of their second intermediate hosts. This occurs in kyphosid fish *Girella laevifrons* infected with metacercariae of the bucephalid *Prosorhynchoides* sp., which encyst mainly in the tail fin [69]. However, the reduction in swimming performance of the fish is not proportional to the intensity of infection, suggesting that, rather than physical interference, the metacercariae may have other effects, such as causing inflammation, localized hemorrhage, or cell destruction, all of which are metabolically demanding [69]. In the grass shrimp *Palaemonetes pugio*, metacercariae of the microphallid *Microphallus turgidus* encyst mainly in the abdominal muscles, and although swimming of the shrimp can be severely affected, this effect is also independent of the intensity of infection [29]. Infected

shrimp spend less time quiescent, more time outside their shelters, and have a shorter delay to first movement after disturbance, making them more easily identifiable by visual predators [31]. In contrast, none of the seven components of the escape response of lobsters varied with grade of infection by *C. solearis*. Not all infected lobsters had cysts in the abdominal muscles, but all except for one had cysts in the coxae of pereopods which, in conjunction with the thoracico-coxal joint, are rich in muscles [70] (see Fig 1). Regardless, cysts did not appear to physically interfere with the tail flips during the retrograde swim, as occurs with Norway lobsters (*Nephrops norvegicus*) infected with the dinoflagellate *Hematodinium* sp. [30]. Therefore, infection with *C. solearis* has no physiological or physical effects on the swimming performance of *P. argus*.

Parasite-induced changes in the concentration of neuromodulators in invertebrate hosts are common [6], and we found increased concentrations of 5-HT in moderately and heavily infected lobsters. In crustaceans, DA modulates activity [71–73] and is associated with increased aggressiveness [74], whereas 5-HT modulates intraspecific agonistic responses as well as swimming and escape behavior [32, 37, 38, 75, 76]. Pérez-Campos et al. [32] observed that most fiddler crabs *Uca spinicarpa* found outside their burrows during the day were infected with cystacanths of the acanthocephalan *Hexaglandula corynosoma*, potentially increasing their susceptibility to predation by herons, the definitive hosts. Infected crabs had a significantly higher concentration of 5-HT in hemolymph, but their concentration of DA did not differ from that of uninfected crabs [32]. We obtained similar results in the present study. The concentration of DA did not differ among our groups of lobsters, but was highly variable in all four groups. In contrast, clear differences occurred in the concentration of 5-HT, with higher values in heavily and moderately infected lobsters than in lightly infected and uninfected lobsters. Higher levels of serotonin might alter the behavior of lobsters, increasing their vulnerability to predators. For example, gammarid amphipods infected with metacercariae of *M. papillorobustus* exhibit an aberrant photophilic behavior [75] that can also be induced by directly injecting them with 5-HT [76, 77]. Subordinate individuals of *Astacus astacus*, *Faxonius virilis* (formerly *Orconectes virilis*), and *Homarus americanus* injected with a solution of 5-HT exhibited a "renovated will" to fight against a dominant conspecific that had previously defeated them [36]. They also exhibited an altered escape response [78] and an increase in fight duration [37, 79], mostly resulting from a decreased likelihood of retreat [80].

Although *P. argus* is mostly nocturnal, it is not uncommon for these lobsters to walk in the open during the day, when encounters with predators are more likely. Upon detecting a nearby predator, exposed lobsters "freeze" to avoid being visually detected. If this fails, then they use their long but stout spiny antennae to fend off attacks, often lunging towards the attacker. Ultimately, they can escape by rapid backwards swimming [50]. The swimming performance of *P. argus* was not affected by infection with *C. solearis*, but heavily infected lobsters had a slightly, but significantly, lower concentration of glucose in the hemolymph. Experiments in vertebrates have revealed that 5-HT might exert different effects on glucose metabolism depending on the species, as well as the hormonal and physiological context (review in [81]). However, the higher concentration of 5-HT in moderately and heavily infected lobsters may cause these lobsters to become more aggressive and attempt to fight an attacker beyond their capabilities, potentially affecting the response of lobsters in predator-prey encounters. It may also affect the social behavior of lobsters, with unknown consequences for casita-based fisheries. A comparison of defense and antipredator behaviors between uninfected and infected lobsters should be conducted in future studies.

In the lobster population of Bahía de la Ascensión, prevalence of infection by *C. solearis* was estimated at 21% and 14% by visual assessment [22, 23], whereas in the lobster population of the Gulf of Batabanó, Cuba, prevalence of infection by *C. solearis* was estimated at ~50% by

full dissection of lobsters [21]. However, all these studies included juvenile lobsters in their estimates. Our results on sensitivity and specificity of visual assessment suggest that for every adult lobster visibly infected with *C. solearis*, there are three additional, not visibly infected adult lobsters, and that there is a 1% chance of confounding marks (such as scars) in muscles with cysts of *C. solearis*. Therefore, visual assessment is insufficient to determine real prevalence of infection with *C. solearis*. We used full dissection as the gold standard, but future studies should aim to develop a less invasive (e.g., molecular) test to determine prevalence of infection [23].

Cruz-Quintana [21] reported encapsulation of cysts of *C. solearis* in some infected lobsters, which is a common immune response in crustaceans [44, 82], but we did not detect any signs of melanization upon full dissection of lobsters, and the total hemocyte count in lobsters was not affected by infection with *C. solearis*. Also, in contrast with other host-parasite systems in which a positive, negative, or non-linear relationship between host size and number of parasites has been found [58, 61], we found no apparent relationship between number of encysted metacercariae of *C. solearis* and adult lobster size. Similarly, Kohler and Poulin [58] found a significant relationship in only 8 out of 21 instances where different species of parasites and crustacean hosts were examined.

In summary, of all the physiological and biochemical characteristics of *P. argus* that we analyzed as a function of infection with *C. solearis*, only two varied with intensity of infection: there was a lower concentration of glucose only in heavily infected lobsters, and a higher serotonin concentration in moderately and heavily infected lobsters relative to lightly infected and uninfected lobsters. These results suggest that infection with *C. solearis* may alter the behavior of spiny lobsters, which might increase the likelihood of trophic transmission of the parasite to the definitive host. Other important host traits that *C. solearis* may affect and warrant further investigation include concentration of other neurotransmitters (e.g., octopamine), female fecundity, and conspecific chemical communication.

## Supporting information

**S1 Table. Response variables of *Panulirus argus* lobsters in four grades of infection by *Cymatocarpus solearis*.** Means and [95% confidence intervals] of all variables compared among four groups of lobsters categorized by the presence and intensity of infection by *C. solearis* (number of metacercarial cysts).
(DOCX)

**S1 Appendix. Full dataset used in the article.** Sheet 1: Variables measured on lobsters (carapace length, total number of cysts, number of cysts in abdomen, number of cysts in cephalothorax, hemolymph components and neurotransmitters, components of the escape response, lobster weight, molt stage). Sheet 2: Number of cysts versus lobster size and sex.
(XLSX)

## Acknowledgments

We thank Edén Magaña-Gallegos for maintaining the seawater tank system and the lobsters well, and for his help in some of the experiments. We also thank Judith Sánchez-Rodríguez, Matteo Cazzanelli, Elisa Y. Chan-Vivas, Charlotte E. Davies, José A. López-Portillo Hurtado, Juan A. Pérez-Vega, Irma Pérez-García and Nancy Herrera-Salvatierra for helping in laboratory activities, and Miguel A. Gómez-Reali, Edgar Escalante-Mancera, and Gustavo Villarreal-Brito for providing technical support. Laura Celis-Gutiérrez helped with literature search.

Water data were provided by Servicio Académico de Monitoreo Meteorológico y Oceanográfico, Puerto Morelos, Q.R.

## Author Contributions

**Conceptualization:** Tomás Franco-Bodek, Rossanna Rodríguez-Canul, Enrique Lozano-Álvarez, Patricia Briones-Fourzán.

**Data curation:** Tomás Franco-Bodek, Cecilia Barradas-Ortiz, Fernando Negrete-Soto, Patricia Briones-Fourzán.

**Formal analysis:** Tomás Franco-Bodek, Patricia Briones-Fourzán.

**Funding acquisition:** Patricia Briones-Fourzán.

**Investigation:** Tomás Franco-Bodek, Cecilia Barradas-Ortiz, Fernando Negrete-Soto, Rossanna Rodríguez-Canul, Enrique Lozano-Álvarez, Patricia Briones-Fourzán.

**Methodology:** Tomás Franco-Bodek, Cecilia Barradas-Ortiz, Fernando Negrete-Soto, Rossanna Rodríguez-Canul, Enrique Lozano-Álvarez, Patricia Briones-Fourzán.

**Project administration:** Fernando Negrete-Soto, Patricia Briones-Fourzán.

**Resources:** Tomás Franco-Bodek, Fernando Negrete-Soto, Rossanna Rodríguez-Canul, Enrique Lozano-Álvarez, Patricia Briones-Fourzán.

**Supervision:** Rossanna Rodríguez-Canul, Patricia Briones-Fourzán.

**Validation:** Tomás Franco-Bodek, Cecilia Barradas-Ortiz.

**Visualization:** Patricia Briones-Fourzán.

**Writing – original draft:** Patricia Briones-Fourzán.

**Writing – review & editing:** Tomás Franco-Bodek, Cecilia Barradas-Ortiz, Fernando Negrete-Soto, Rossanna Rodríguez-Canul, Enrique Lozano-Álvarez, Patricia Briones-Fourzán.

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
