## [Decision Letter · Decision Letter 0]

24 Feb 2023

PONE-D-23-02958Effects of Cymatocarpus solearis (Trematoda: Brachycoeliidae) on its second intermediate host, the Caribbean spiny lobster Panulirus argusPLOS ONE

Dear Dr. Briones-Fourzán,

Thank you for submitting your manuscript to PLOS ONE. After careful consideration, we feel that it has merit but does not fully meet PLOS ONE’s publication criteria as it currently stands. Therefore, we invite you to submit a revised version of the manuscript that addresses the points raised during the review process.

We look forward to receiving your revised manuscript.

Kind regards,

Hudson Alves Pinto, Ph.D

Academic Editor

PLOS ONE

Journal Requirements:

Additional Editor Comments:

Dear Authors,

Your MS received good reports from both Reviewers, and I think it has potential to be accepted after careful analysis of the corrections.

Best regards,

Hudson

Reviewers' comments:

Reviewer's Responses to Questions

**Comments to the Author**

1. Is the manuscript technically sound, and do the data support the conclusions?

Reviewer #1: Yes

Reviewer #2: Yes

2. Has the statistical analysis been performed appropriately and rigorously? 

Reviewer #1: No

Reviewer #2: Yes

3. Have the authors made all data underlying the findings in their manuscript fully available?

Reviewer #1: Yes

Reviewer #2: No

4. Is the manuscript presented in an intelligible fashion and written in standard English?

Reviewer #1: Yes

Reviewer #2: Yes

5. Review Comments to the Author

Reviewer #1: I have uploaded my full review as an attachment.

I believe the manuscript meets all PLOS ONE publication criteria. The work is scientifically sound, and I have no concerns about the experimental design. However, I would suggest that the authors implement some additional statistical tests to take full advantage of the data that they have painstakingly collected. For the most part, the authors limited themselves to using non-parametric hypothesis tests (Kruskal-Wallis) to test for significant differences. These tests can be useful, but they must be interpreted carefully because of problems with statistical vs. biological significance (i.e., Johnson 1995). More to the point, these tests have no real predictive abilities. Given the wealth of data collected for this project, I think the manuscript could be improved using glms and/or mixed models. These analyses would address the question: which variable(s) account for most of the variation in infection intensity? As a reader, I was left wondering about this. I have more detailed suggestions along these lines in the Methods/Results section of this review.

Reviewer #2: The authors didn't mention making their data available, but they might have somewhere and mention it when they submitted their work

General Comments:

Overall, I think this is a very informative study that adds to the current understanding of how parasites can alter crustaceans behavior and physiology, specifically, and the potential ecological and economic ramifications of parasites, at large. While I think my comments can greatly aid in the readability of this document, they are admittedly all relatively minor, as I think the authors work stands on its own.

Abstract

• In line 30, change “their” to the hosts’

• Split lines 34 – 38 into 2 sentences, one about the physiological parameters studied and one about the behavioral assays conducted

• Line 45- boldness and aggression are specific behavioral personalities that can be shown with repeated testing. It seems strange to include those terms here considering this paper isn’t about personalities

• Move the sentence in lines 44 – 45 up to right before the results

• Lines 44 – 47 don’t seem to fit in the abstract, or if the author feels they must be included, should be moved up to the methods part of the abstract instead of as the last sentence

Introduction:

• In general, the flow of the introduction seems a bit off. It immediately gives information about the host and parasite then go back to talking about parasite infection at large. All of the information is good, it’s just the order. Also, including transition sentences as the last sentence in each paragraph would greatly aid in flow for the reader

• Add a sentence at the end of the first paragraph to also include behavioral impact of parasites on hosts because this study includes physiological and behavioral impacts

• I found some studies (e.g. Gnanalingam and Bulter 2018) that say P. argus also eat gastropods, which could be important for parasite transmission, so should be included in their list of prey items in line 73

• Lines 75-78 don’t seem relevant to the paper, unless the parasite can be transmitted between 2 lobsters, which seems very unlikely

Materials and Methods:

• Lines 114-116, depth of tanks?

• Line 117- is there a known range for salinity and temperature during the time of the experiments that could be included here?

• Lines 124-126, include Ns for each

• Line 128, what is the N?

• Line 136 – 137- how were the cysts quantified?

• This response is called a tail flip in other crustaceans. Is it called that in lobsters?

• Were the lobsters used in these experiments fully intact?

• Lines 175 – 182 read like introduction material, not materials and methods, and should be moved accordingly. The information is pertinent to the paper

• Lines 212-214- the hepatopancreas also acts as a filter, removing toxins from the hemolymph

• Lines 218-227 also read as introduction material and should be moved

• Line 241 mentions these categories rendered samples of similar size. Does this mean the lobsters were of similar size or there was a similar N for all categories?

• No normality results are reported, so why was a Kruskal-Wallis test used?

• If lobsters were all dissected and metacercariae counted post-mortem anyway, why not just use these data for all analyses instead of using the visual tests of alive lobsters at all?

Results:

• While I have no doubt that the statistical analyses showed significant results for some of the escape response, figure 2 looks like there are no differences. Would bar graphs with error bars be more informative here?

• Same comment for figure 3

Discussion

• Lots of specific anatomical terminology is used, especially in lines 396-409. This terminology is definitely relevant to the findings, but adding another figure that is a labeled diagram of a lobster would greatly aid in readers’ understanding, especially if not familiar with crustacean anatomy

• Adding transition sentences throughout the discussion would also aid in flow, similar to the introduction

• Did that authors relate the fiddler crab behavior with 5-HT concentration for the example mentioned in 427-431? If so, that should be mentioned here because as written, this seems like 2 unrelated findings

• Line 439- Orconectes virilis has been reclassified as Faxonius virilis

6. PLOS authors have the option to publish the peer review history of their article (what does this mean?). If published, this will include your full peer review and any attached files.

Reviewer #1: No

Reviewer #2: No

---

## [Author Response · Author response to Decision Letter 0]

1 May 2023

Manuscript Number: PONE-D-23-02958

Manuscript Title: Effects of Cymatocarpus solearis (Trematoda: Brachycoeliidae) on its second intermediate host, the Caribbean spiny lobster Panulirus argus

Dear Editor: Our responses to reviewers’ comments are in blue.

Journal Requirements:

R: Grant number has been corrected.

R. Caption for Supporting Information has been added at the end of the manuscript. In-text citation matches (Line 249 of revised manuscript).

Reviewer 1.

Digenean trematode life cycles are fascinating and wonderfully complex. Many interesting ecological questions can be tested at the intermediate host-stage, as this is where behavioral manipulation of hosts is theorized to occur. In this paper, the authors quantified the effects of C. solearis infection on the Caribbean spiny lobster P. argus, a second-intermediate host for the parasite. There were clear differences in biomarkers of infection (e.g., concentrations of glucose and serotonin) and physical performance indicators (e.g., escape response) as a function of infection intensity in P. argus, findings that increase the likelihood of trophic transmission to an appropriate definitive host. 

I believe the manuscript meets all PLOS ONE publication criteria. The work is scientifically sound, and I have no concerns about the experimental design. However, I would suggest that the authors implement some additional statistical tests to take full advantage of the data that they have painstakingly collected. For the most part, the authors limited themselves to using non-parametric hypothesis tests (Kruskal-Wallis) to test for significant differences. These tests can be useful, but they must be interpreted carefully because of problems with statistical vs. biological significance (i.e., Johnson 1995). More to the point, these tests have no real predictive abilities. Given the wealth of data collected for this project, I think the manuscript could be improved using glms and/or mixed models. These analyses would address the question: which variable(s) account for most of the variation in infection intensity? As a reader, I was left wondering about this. I have more detailed suggestions along these lines in the Methods/Results section of this review. 

• Johnson, D. H. (1995). Statistical sirens: the allure of nonparametrics. Ecology, 76(6), 1998-2000.

R1: We appreciate the suggestion of using parametric analyses for the data. We have now used general linear models (GLM), i.e., conventional linear regression models for a continuous response variable given continuous and/or categorical predictors. We were worried about violating some assumptions of parametric tests, but this was not an issue since we used robust tests. Please see our response no. 7 (R7) below.

Introduction

Lines 79-89 -- why is visual assessment important? I think some additional context here would help. Is it just for researchers interested in the presence of the parasite, or do fishermen care as well? If appearance is important, this may be something to return to in the Discussion (beginning around line 460). 

R2: If visual assessment of infection prevalence provided a good approximation to real prevalence, it would be a rapid, non-invasive way of assessing infection in the lobster populations, which would be of interest to both researchers and fishers. However, here we prove that visual assessment misses many infected lobsters and hence cannot be used for rapid assessments.

Lines 89-90 – the transition to the next paragraph about trophic transmission seemed a little abrupt. Perhaps another sentence is needed here?

R3. The Introduction has been completely rearranged to improve transition between paragraphs.

Lines 98-102 – The ideas here seemed a little vague. I think the reader may be left wondering why these parasites have more consequences for intermediate hosts. The intermediate host needs to be eaten for the parasite to complete its lifecycle in a definitive host. Perhaps this should be stated? Also, how would parasites affect the lobster fishery? 

R4. Thank you for this suggestion. The paragraph has been revised.

Methods/Results

Line 182 – consider citing (and perhaps later discussing?) Shaw et al. 2009: 

• Shaw, J. C., Korzan, W. J., Carpenter, R. E., Kuris, A. M., Lafferty, K. D., Summers, C. H., & Øverli, Ø. (2009). Parasite manipulation of brain monoamines in California killifish (Fundulus parvipinnis) by the trematode Euhaplorchis californiensis. Proceedings of the Royal Society B: Biological Sciences, 276 (1659), 1137-1146.

R5: Thanks for recommending this paper, which is relevant to our manuscript. 

Suggestions for additional post-hoc testing: 

It is possible to have a low p-value but a small effect size. In other words, the effect detected by the test that seemed so important (i.e., with the low p-value) is actually quite small. Cumming (2012) gives an excellent overview of this issue. To provide some additional context for your results from the Kruskal-Wallis tests, I might suggest testing the magnitude of effect for each significant p-value. The package “rstatix” is quite useful for basic statistical tests – I think Cohen’s D is preferred for parametric data and Wilcoxon effect size tests for non-parametric data. 

• Cumming G. 2012. Understanding The New Statistics: Effect Sizes, Confidence Intervals, and Meta-Analysis (Multivariate Applications Series)

R6: After checking major assumptions for parametric tests, we have changed all statistical analyses to GLMs and do not use Kruskal Wallis tests in the revised manuscript. 

Suggestions for model building using information theoretic approaches: 

You probably have your own methods, but in the interests of being as helpful as possible I’ll share with you what packages I typically use for multimodel inference in R. 

• glmmTMB – package for building and testing generalized linear mixed models that has a great suite of diagnostic tools for assessing model fit: 

https://cran.r-project.org/web/packages/glmmTMB/index.html

• DHARMa – add-on package with helpful tests for assisting with model convergence issues:

https://cran.r-project.org/web/packages/DHARMa/vignettes/DHARMa.html

• AICcmodavg – package for ranking and comparing models using information theoretic approaches: 

https://cran.r-project.org/web/packages/AICcmodavg/index.html

You could bin the data into infected (1 or more cysts) and uninfected (0 cysts) categories. The goal here would be to rank and compare models testing the likelihood of infection. What variable(s) best predict whether a lobster is infected or not? Your response variable would be infection status (yes or no), a binary outcome (1 or 0), and you would fit the data to a binomial error distribution. In models that incorporate swimming performance as a predictor, you could consider adding trial number as a random effect as this might account for some of the variation in your response variable but would not be of biological/ecological importance. 

You could also build models using infection intensity as your response variable (0 cysts; 1-10 cysts; 11-30 cysts; > 30 cysts). What variable(s) predict the likelihood of infection intensity in each case? You would fit the data to a negative binomial distribution if they showed evidence of overdispersion, which may be the case for the final category – otherwise, a Poisson would probably fit the data better. For variables that had multiple trials, consider using trial number as a random effect in your model structure. 

R7. Thank you for all this interesting and very useful information. However, in our study we were not trying to find variables predicting whether a lobster was infected or not, or using infection intensity as the response variable. Quite the opposite: we were trying to find how infection intensity (the predictor) affected a number of response variables of the lobsters (i.e., components of the escape response, growth rate, concentration of metabolites or biogenic amines, etc.). Originally, we had planned on measuring the entire group of variables in the same lobsters, but this was not possible due to time constraints, logistic problems, death of several lobsters, molting of lobsters, incompatibility of some experiments, recording errors, etc. However, it is true that parametric tests are more powerful, and many are robust to departure from some assumptions. Therefore, following your advice, we have now changed all statistical analyses. We used multivariate analyses of variance (MANOVA) with a general lineal model (GLM) approach to test for potential effects of parasite intensity (with four levels: uninfected, lightly infected, moderately infected, and heavily infected) on two groups of variables that were measured in the same lobsters: (a) the components of the escape response, and (b) the concentration of metabolites in hemolymph. In (b), all MANOVA assumptions were met, but in (a), the assumption of homogeneity of the variance-covariance matrix was not met. However, MANOVA is robust to departure from this assumption when sample sizes are equal or nearly equal (which was the case) (Tabacknik & Fidell 2012; Nimon 2012). The rest of the variables (concentration of dopamine, concentration of serotonin, hemocyte count, hepatosomatic index, and growth rate) were individually subjected to ANCOVAs, using size as a covariate.

Upon doing the MANOVA for the components of the escape response, I noticed an unfortunate mistake in the calculation of acceleration (m/s/s) in the group of uninfected lobsters due to the use of an erroneous column in the Excell sheet. The spurious results had led us to conclude that there were significant differences in acceleration (and in work performed and force exerted, which are calculated from acceleration) of uninfected lobsters relative to the three groups of infected lobsters, when in fact the differences were not significant. In view of this mistake, we double-checked again all data, and everything else was correct. Thus, although acceleration did not differ among lobster groups, these groups did differ significantly in the concentration of serotonin and glucose in the hemolymph.

Tabachnick BG, Fidell LS (2012) Using Multivariate Statistics, 6th edn. Pearson, London.

Nimon KF (2012) Statistical assumptions of substantive analyses across the general linear model: a mini-review. Front. Psychol. 3: 322.

Some thoughts on lobster size as a predictor variable:

I think you could make an argument for using lobster size as a fixed effect (predictor) in your models, or as a random effect since there doesn’t seem to be a clear linear relationship between size and infection status (lines 280-281). Admittedly, this result is a little surprising since larger (older) organisms tend to have more cysts, which you mention (lines 248-249). However, it may also be possible for crustaceans to “clear” their cysts over time. Older cysts can appear brown and withered (Blakeslee et al. 2020), a result of the long-term crustacean innate immune response (Lee and Soderhall 2002); however, you did not find evidence of melanization (line 472). One other intriguing possibility for this non-linear relationship is that larger adults are foraging in different areas compared to smaller individuals and so are not in proximity as often to the first-intermediate host. I have some unpublished data from stone crabs Menippe mercanaria that support this idea. Smaller individuals (up to approx. 40 mm carapace width) are sympatric with Panopeid mudcrabs in shallow water oyster reefs. These individuals often have trematode cysts. Larger stone crabs, on the other hand, tend to burrow more and seek out deeper habitats, meaning that they are not as close on average to the first-intermediate host, the mudsnail Ilyanassa obsoleta. I rarely find cysts in these larger stone crabs. 

All in all, since the biological relationship between size and infection status seems unclear, I might argue for using size as a random effect since it is not an important predictor of infection but may still account for some variation in your response variables. 

• Blakeslee, A., Ruocchio, M., & Moore, C. S. (2020). Altered susceptibility to trematode infection in native versus introduced populations of the European green crab.

• Lee, S. Y., & Söderhäll, K. (2002). Early events in crustacean innate immunity. Fish & shellfish immunology, 12(5), 421-437.

R8. In lines 249-249 of previous ms we were citing Koehler & Poulin (2010); ours is the first estimation of number of C. solearis cysts in P. argus lobsters. We found no relationship between lobster size and number of cysts by full dissection, so a larger number of cysts in larger organisms does not appear to apply to this system. Regardless, we included size as a covariate in the GLMs that we performed for the variables that were measured independently. We did this because most statistics books concur in that continuous variables such as size cannot be used as random effects but as covariates. 

Discussion

I’ll mostly hold off on commenting on this section for now, as your results may change if you decide to incorporate my suggestions for additional analyses. Ultimately, I think what you have is great, but I think you could do a lot more with your data. 

R9. Thank you very much for all your comments. The Discussion has indeed changed, mostly because of the detected mistake in the estimation of acceleration (and hence in force and work, which are derived from acceleration) for uninfected lobsters. Upon correcting the data, we found no significant effect of parasite intensity on any component of the escape response, and the Discussion was changed accordingly. 

One note on the serotonin (5-HT) results: 

Would elevated levels of 5-HT affect lobsters’ ability, or interest, to shelter together in casitas? How would this affect the fishery? 

R10. It might, since in other decapods higher levels of 5-HT increase their “will to fight”. But whether this would affect the fishery remains to be determined. Therefore, we modified the Discussion like this (new Lines 464-469): “the higher concentration of 5-HT in moderately and heavily infected lobsters may cause these lobsters to become more aggressive and attempt to fight an attacker beyond their capabilities, potentially affecting the response of lobsters in predator-prey encounters. It may also affect the social behavior of lobsters, with unknown consequences for casita-based fisheries. A comparison of defense and antipredator behaviors between uninfected and infected lobsters should be conducted in future studies.”

WE THANK REVIEWER 1 FOR THE THOROUGH REVIEW AND HELPFUL COMMENTS.

Reviewer 2

General Comments:

 Overall, I think this is a very informative study that adds to the current understanding of how parasites can alter crustaceans behavior and physiology, specifically, and the potential ecological and economic ramifications of parasites, at large. While I think my comments can greatly aid in the readability of this document, they are admittedly all relatively minor, as I think the authors work stands on its own. 

R. We appreciate the positive feedback.

Abstract

• In line 30, change “their” to the hosts’

R. Done.

• Split lines 34 – 38 into 2 sentences, one about the physiological parameters studied and one about the behavioral assays conducted

R. Done.

• Line 45- boldness and aggression are specific behavioral personalities that can be shown with repeated testing. It seems strange to include those terms here considering this paper isn’t about personalities 

R. The terms bolder and more aggressive have been deleted from the Abstract.

• Move the sentence in lines 44 – 45 up to right before the results

R. Done.

• Lines 44 – 47 don’t seem to fit in the abstract, or if the author feels they must be included, should be moved up to the methods part of the abstract instead of as the last sentence.

R. These lines were removed from the Abstract.

Introduction:

• In general, the flow of the introduction seems a bit off. It immediately gives information about the host and parasite then go back to talking about parasite infection at large. All of the information is good, it’s just the order. Also, including transition sentences as the last sentence in each paragraph would greatly aid in flow for the reader

R. Thank you for your suggestions. The Introduction has been extensively rearranged.

• Add a sentence at the end of the first paragraph to also include behavioral impact of parasites on hosts because this study includes physiological and behavioral impacts

R. Thanks for the suggestion.

• I found some studies (e.g. Gnanalingam and Bulter 2018) that say P. argus also eat gastropods, which could be important for parasite transmission, so should be included in their list of prey items in line 73

R. The diet of these lobsters is opportunistic and very diverse. They eat bivalves, gastropods, chitons (i.e. mollusks), cirripedes and all kinds of decapods, (crustaceans), ophiurids, urchins, sea cucumbers (echinoderms) and all kinds of annelids (worms). Because of this and because there is no trophic transmission of these trematodes from the first to the second intermediate host, we only mentioned the higher taxa. 

• Lines 75-78 don’t seem relevant to the paper, unless the parasite can be transmitted between 2 lobsters, which seems very unlikely

R. These lines were intended to explain why casitas are used in some fisheries, such as the one from which we obtained the lobsters, and because, as suggested by Reviewer 1, the increase in serotonin in infected lobsters may alter their social behavior, which is now addressed in the Discussion.

Materials and Methods:

• Lines 114-116, depth of tanks?

R. Depth of tanks has been added.

• Line 117- is there a known range for salinity and temperature during the time of the experiments that could be included here?

R. Yes. The ranges have been added.

• Lines 124-126, include Ns for each

R. Done. 

• Line 128, what is the N?

R. The sample size has been added. 

• Line 136 – 137- how were the cysts quantified?

R. All cysts found in a lobster upon full dissection were counted.

• This response is called a tail flip in other crustaceans. Is it called that in lobsters?

R. Yes. This name has been added.

• Were the lobsters used in these experiments fully intact?

R. Yes, except for 4, which were lacking 1 appendix (leg or antenna), and 2 which lost it during the experiment.

• Lines 175 – 182 read like introduction material, not materials and methods, and should be moved accordingly. The information is pertinent to the paper

R. These lines were added here as an explanation of why we considered important to compare that characteristic. However, following your advice, we have moved a shortened version of all this material (and similar parts related to other characteristics) to the last part of the Introduction.

• Lines 212-214- the hepatopancreas also acts as a filter, removing toxins from the hemolymph

R. We have removed this piece of information.

• Lines 218-227 also read as introduction material and should be moved.

R. Same response as to Lines 175-182.

• Line 241 mentions these categories rendered samples of similar size. Does this mean the lobsters were of similar size or there was a similar N for all categories?

R. It means a similar N for all lobster groups. Sentence has been rephrased.

• No normality results are reported, so why was a Kruskal-Wallis test used?

R. We have changed all the statistical analyses to parametric tests. We now used MANOVAs and ANOVAs with a General Lineal Model (GLM) approach.

• If lobsters were all dissected and metacercariae counted post-mortem anyway, why not just use these data for all analyses instead of using the visual tests of alive lobsters at all?

R. In previous studies, prevalence of infection by C. solearis in wild lobsters was estimated only by visual assessment. We wanted to test how “accurate” these visual assessments are, because they could be used to conduct rapid assessments elsewhere. Therefore, we conducted the validity measures but find out that visual assessment missed many infected lobsters, so it is not a good method for rapid assessments.

Results:

• While I have no doubt that the statistical analyses showed significant results for some of the escape response, figure 2 looks like there are no differences. Would bar graphs with error bars be more informative here?

R. These graphs have been changed because we found a mistake in the data processing, for which we apologize. Acceleration had been wrongly estimated (see response R7 to Reviewer 1). After correcting the mistake, no significant differences were found in any components of the escape response. Analyses and figures have now been corrected. However, we believe that, unlike column graphs with error bars, box plots show the actual distribution of the raw data, so they are a good choice when comparing groups.

• Same comment for figure 3.

R. Please see previous response.

Discussion

• Lots of specific anatomical terminology is used, especially in lines 396-409. This terminology is definitely relevant to the findings, but adding another figure that is a labeled diagram of a lobster would greatly aid in readers’ understanding, especially if not familiar with crustacean anatomy

R. Thanks for this suggestion. We have added a photo of a lobster with the relevant names (Fig. 1 in revised ms).

• Adding transition sentences throughout the discussion would also aid in flow, similar to the introduction

R. The discussion has been changed to address the amended results. Care has been taken to enhance the flow between paragraphs.

• Did that authors relate the fiddler crab behavior with 5-HT concentration for the example mentioned in 427-431? If so, that should be mentioned here because as written, this seems like 2 unrelated findings.

R. Yes, it was the same authors. The sentence has been rephrased to improve clarity.

• Line 439- Orconectes virilis has been reclassified as Faxonius virilis

R. Thanks for catching this. The name has been corrected. 

WE THANK REVIEWER 2 FOR THE THOROUGH REVIEW AND HELPFUL COMMENTS.

---

## [Decision Letter · Decision Letter 1]

18 May 2023

PONE-D-23-02958R1Effects of Cymatocarpus solearis (Trematoda: Brachycoeliidae) on its second intermediate host, the Caribbean spiny lobster Panulirus argusPLOS ONE

Dear Dr. Briones-Fourzán,

Thank you for submitting your manuscript to PLOS ONE. After careful consideration, we feel that it has merit but does not fully meet PLOS ONE’s publication criteria as it currently stands. Therefore, we invite you to submit a revised version of the manuscript that addresses the points raised during the review process.

We look forward to receiving your revised manuscript.

Kind regards,

Hudson Alves Pinto, Ph.D

Academic Editor

PLOS ONE

Journal Requirements:

Additional Editor Comments:

I congrats the authors for the excelent revision.

Some final interesing comments are presented by Reviewer 1.

Reviewers' comments:

Reviewer's Responses to Questions

**Comments to the Author**

1. If the authors have adequately addressed your comments raised in a previous round of review and you feel that this manuscript is now acceptable for publication, you may indicate that here to bypass the “Comments to the Author” section, enter your conflict of interest statement in the “Confidential to Editor” section, and submit your "Accept" recommendation.

Reviewer #1: All comments have been addressed

2. Is the manuscript technically sound, and do the data support the conclusions?

Reviewer #1: Yes

3. Has the statistical analysis been performed appropriately and rigorously? 

Reviewer #1: Yes

4. Have the authors made all data underlying the findings in their manuscript fully available?

Reviewer #1: Yes

5. Is the manuscript presented in an intelligible fashion and written in standard English?

Reviewer #1: Yes

6. Review Comments to the Author

Reviewer #1: (No Response)

7. PLOS authors have the option to publish the peer review history of their article (what does this mean?). If published, this will include your full peer review and any attached files.

Reviewer #1: No

---

## [Author Response · Author response to Decision Letter 1]

25 May 2023

Manuscript Number: PONE-D-23-02958

Manuscript Title: Effects of Cymatocarpus solearis (Trematoda: Brachycoeliidae) on its second intermediate host, the Caribbean spiny lobster Panulirus argus

Response to authors’ revisions: 

I have read the authors’ response-to-reviewers document and the revised manuscript, which overall is much improved. The authors have done a thorough job with their revisions. In the process of reanalyzing the data, they uncovered and corrected a typo in the original dataset that would have affected their results and discussion. They also switched their primary method of analysis from hypothesis testing (i.e., Kruskal-Wallis) to regression-based methods, which are more comprehensive and informative. Parts of the Introduction and Discussion were reorganized as needed based on feedback from both reviewers. 

At this point, I have only minor requests for clarification, primarily with how the stats are reported. I am trying to be as helpful as possible, as I think the work is interesting and very informative! 

R. We appreciate the positive feedback.

• Methods

You mention that you used general linear models (line 314) as opposed to generalized linear models. The former assumes that the residuals from your dependent variable are normally distributed while the latter does not. Parasite count data is often non-normal…I believe your dependent variable was parasite intensity? (line 315). I think it would help to clarify that the residuals of your data were normally distributed and then report that prior to discussing how you used GLMs. Otherwise, you should switch to using generalized linear models, which are preferred for modeling data with non-normal residuals.

R. As previously explained, parasite intensity was not the dependent variable; it was the factor (independent variable) whose effect was tested on several response variables. That is, parasite intensity was considered as a categorical factor with four levels: uninfected, lightly infected, moderately infected, and heavily infected lobsters. A GLM MANOVA tested the effect of this factor (parasite intensity) on a combination of the four metabolites (albumin, cholesterol, glucose, and total protein) measured in the hemolymph of lobsters on the one side. Another tested the effect of parasite intensity on a combination of the seven components of the escape response of lobsters on the other side. The rest of the variables were individually compared between lobster groups with separate GLMs.

If you stick with the general linear models, I might suggest reporting a few other test statistics for your ANCOVA results. By themselves, p-values convey very little information about the magnitude of effect, i.e., how powerful, or significant, your results are. I would suggest including information on marginal means and their associated 95% confidence intervals and reporting this for each significant effect in your results. Here’s some additional information on reporting CIs and marginal means for ANCOVAs: https://www.medcalc.org/manual/analysis-of-covariance.php

R. Thanks for this suggestion. We have added a large supplementary table (S1 Table) with the means and their 95% confidence intervals for all the response variables that we measured and compared among the four groups of lobsters (uninfected, lightly infected, moderately infected, heavily infected). 

• Other

I think it would help to have a final concluding sentence in your Abstract that kind of ties things together, a more general statement about how your results apply more broadly. You have something like this in your conclusion section (p. 29) around lines 626-629.

R. OK. We have changed the final sentence of the Abstract to the following: “As changes in 5-HT concentration can modify the host’s activity patterns or choice of microhabitat, our results suggest that infection with C. solearis may alter the behavior of spiny lobsters, potentially increasing the likelihood of trophic transmission of the parasite to the definitive host.”

---

## [Editor Report · Decision Letter 2]

30 May 2023

Effects of Cymatocarpus solearis (Trematoda: Brachycoeliidae) on its second intermediate host, the Caribbean spiny lobster Panulirus argus

PONE-D-23-02958R2

Dear Dr. Briones-Fourzán,

We’re pleased to inform you that your manuscript has been judged scientifically suitable for publication and will be formally accepted for publication once it meets all outstanding technical requirements.

Kind regards,

Hudson Alves Pinto, Ph.D

Academic Editor

PLOS ONE

Additional Editor Comments (optional):

I congrats the Authors for their efforts in the revision. The MS is now suitable for publication and will be is an important cientific contribution for the field.

---

## [Editor Report · Acceptance letter]

1 Jun 2023

PONE-D-23-02958R2 

Effects of <i>Cymatocarpus solearis<i/> (Trematoda: Brachycoeliidae) on its second intermediate host, the Caribbean spiny lobster *Panulirus argus*

Dear Dr. Briones-Fourzán:

I'm pleased to inform you that your manuscript has been deemed suitable for publication in PLOS ONE. Congratulations! Your manuscript is now with our production department. 

Kind regards, 

on behalf of

Dr. Hudson Alves Pinto 

Academic Editor

PLOS ONE